# Designing Urban Green Space (UGS) to Enhance Health: A Methodology

**DOI:** 10.3390/ijerph17145205

**Published:** 2020-07-18

**Authors:** Esther J. Veen, E. Dinand Ekkel, Milan R. Hansma, Anke G. M. de Vrieze

**Affiliations:** 1Department of Social Sciences, Rural Sociology, Wageningen University and Research, Hollandseweg 1, 6706KN Wageningen, The Netherlands; anke.devrieze@wur.nl; 2Aeres University of Applied Science, Stadhuisstraat 18, 1315HC Almere, The Netherlands; d.ekkel@aeres.nl (E.D.E.); milanhansma@gmail.com (M.R.H.)

**Keywords:** urban green space, health benefits, urban design, evidence-based design

## Abstract

Policymakers and urban designers strive to implement the increasing evidence about the positive association between urban green space (UGS) and health in policy. In Almere, The Netherlands, the Regenboogbuurt (“Rainbow Quarter”) neighbourhood is currently being revitalized. The research team was asked to deliver design principles for the improvement of UGS in this neighbourhood to benefit the health of its residents. However, robust studies that demonstrate what UGS criteria offer what particular benefit for what target group are scarce. This paper contributes to the need for more evidence-based UGS design by presenting the approach we used to develop UGS design principles for Regenboogbuurt. Demographic information, health statistics, residents’ opinions, and data about the current use of UGS were analysed to choose target groups and to formulate health benefit goals. We also developed a model for assessing the health benefits of UGS. For two age groups (those aged 10–24 and 40–60), stimulating physical health and social cohesion, respectively, were determined to be the goals of improving UGS. UGS design principles were then assessed based on the existing literature. These principles will be taken into account when this area is revitalized in 2021. Thus, there will be an opportunity to measure whether these design principles did indeed contribute to residents’ health.

## 1. Introduction

The share of the world’s population living in cities is projected to increase from 54% in 2015 to 66% by 2050 [1]. As cities grow, they increasingly face environmental challenges: urban green space (UGS) can play an important role in mitigating them [2]. Moreover, there is substantial evidence that UGS is positively associated with the health of urban citizens [3,4,5,6] (for a recent systematic review, see [6]). UGS is therefore of great value for ensuring socio-environmentally sustainable urban growth [7]. However, while policymakers and urban planners are keen to commission experts to design UGS in such a way that it produces health benefits for certain populations [8], thus far, UGS has hardly been designed to provide any specific health benefits. One of the reasons for this is that it is far from clear what characteristics of UGS offer what particular benefit to what particular group. Longitudinal studies in which such criteria are developed and applied, and their effects measured, are rare and of poor quality [9]. Moreover, the effects that completed UGS projects have on public health often go unevaluated, and therefore, it frequently remains unclear whether these projects achieve their stated goals [10]. Several authors [9,10,11] therefore call for more robust, evidence-based research to increase the knowledge base about how certain aspects of UGS affect health. Brown and Corry [10] make a case for “evidence-based landscape architecture”, arguing that scholarly evidence needs to be used more explicitly and deliberately, and that landscape architecture should be based on facts rather than beliefs. Two recent papers [9,12] reviewing evidence on the health benefits of built environment interventions emphasize the paucity of robust evidence related to urban environmental interventions. Benton et al. [13] made a similar observation in their review paper about the effects of changing the built environment on physical activity. 

Besides a lack of rigorous scientific studies, the complexity of designing for the real world makes it difficult to apply scarce scientific knowledge. We identified five factors that contribute to this complexity. First, the contextuality of green space and its users makes it difficult to create actual transferable evidence. Hence, although Lee et al. [8] (p. 134) stress the need to identify “what health outcomes are sought, what activities in urban green spaces contribute to these outcomes, and in turn identify what features of urban green space would encourage such activities”, the contextual reality may often not be as straightforward. Second, UGS provides various interrelated health benefits that are hard to disentangle, and mutually influence each other. For instance, being physically active generally improves mental health, but may also lead to social cohesion when people spend time outdoors and meet others, which in turn may positively influence mental health. These relationships will be discussed in more detail in the next section. Third, UGS may bring different benefits to different layers of society, and therefore, those who design UGS must consider the needs of different target groups [14]. For example, young people may lack physical exercise, while senior citizens may be in need of social cohesion. Moreover, similar needs in different groups may be best targeted in different ways (i.e., physical exercise is stimulated differently in children and adults). Fourth, “urban green space” is a broad concept: it refers to “vegetated areas located within built-up areas, including natural and planted trees, grass, shrubs and flowers” [15] (p. 578) and may also include urban forests [16]. Green spaces can thus come in many forms and sizes and incorporate various attributes or elements (i.e., benches, flower beds, trees). Fifth, other (environmental) features affecting the use of UGS, and thus influencing potential health benefits, are relevant as well. Examples are cleanliness, accessibility, attractiveness, perceived safety and security, and distance from potential users’ homes [8]. Perceived safety is often recognized as an important factor influencing the use of UGS (see, e.g., [4,8,17]). These five factors that contribute to the complexity of designing for the real world show that “naïve assumptions of simple cause and effect relationships are unhelpful from a planning perspective” [18] (p. 781): designing UGS to improve public health is complicated.

Although there is a pressing need to gather more evidence on the relation between specific UGS designs and health, related contextual aspects must be accounted for as well. Rather than evidence-based design in the strict sense of the word (i.e., an approach in which only measures that have been “scientifically proven” to work in all circumstances are applied), we advocate gathering “contextual evidence” through intervention-based studies that review the effects of UGS interventions on health. We argue, moreover, that this approach requires a very careful, robust, and transparent process from the outset; an initial baseline study must be conducted to enable a thorough evaluation of the interventions after they have been implemented. Hence, we believe that it is most important to carefully consider designs for green spaces by not “merely” relying on evidence created in other projects but also by applying a rigorous design process in which the specificities of the neighbourhood are taken into account.

In this paper, we present an approach to developing such a robust design process that takes the contextual background into consideration. This approach is based on available evidence and takes into account the local context, while simultaneously generating possibilities to produce evidence in the future. We present our approach by using the neighbourhood Regenboogbuurt, in Almere, The Netherlands, as a case study. In 2018, the Almere municipal government began working on revitalizing Regenboogbuurt, which will continue until 2021. The objective of the revitalizations is to deal with problems caused by land subsidence; the streets will be elevated and public spaces redesigned to increase the quality of the living environment. Public green spaces in the neighbourhood will also be improved. Our research team was asked to develop design principles for these green spaces that will substantially contribute to the health and wellbeing of Regenboogbuurt’s residents. (We understand “health and wellbeing benefits” as effects that improve people’s physical and mental health, including social relations. As we consider wellbeing to be an element of health, in the rest of this paper we refer only to “health benefits”.) These planned improvements make for an interesting “before” and “after” study. They enable conducting a baseline study and a follow-up study after the UGS has been redesigned and realized in order to analyse its impact on health. The value of such an evaluation lies in improving insights into what can work in a certain situation (and what cannot).

In sum, the aim of this paper is to justify the approach we used to propose design principles for Regenboogbuurt, and, more importantly, to inform planners and designers about this approach and share with them the steps that can be taken to incorporate considerations about a neighbourhood’s unique features when designing UGS to promote public health. An important element of our approach is the use of demographic information, health statistics, residents’ opinions, and data about current use of UGS to target the design to the neighbourhood’s population. By applying this strategy, we acknowledge that the ways in which green space can improve public health are highly context dependent. The novelty of our paper lies in this approach to, or methodology for, designing UGS, rather than in the concrete results of the case study presented (an evaluation of the interventions based on our design principles is planned for a later date). Nevertheless, the case study is an essential element of the paper, as it allows a departure from a more generic level of explanation and enables us to demonstrate our approach in a real-life practical setting. We see this as vital to our argument as it is exactly such practice-based approaches that policymakers request and which are currently lacking in the literature. In other words, this paper is not a response to the lack of knowledge on the effects of UGS on health as such, but rather to the difficulty to apply this knowledge, and it suggests how to do it so that more knowledge can be created. 

As a consequence of our somewhat unusual starting point (i.e., presenting an approach that is simultaneously applied in a case study), the structure of this paper deviates from the standard: it asks the reader to accompany us on the journey through applying our approach. The first step in our approach is defining the potential health benefits of UGS and how they can be effected: this is explained in the next section. In the third section we present our baseline study, the second step, clarifying how we took Regenboogbuurt’s specific features into account. Thereafter, we explain the third step in our approach: selecting the target groups and the health benefits that our design principles concentrate on. We then, in the final step, translate these findings into design principles, based on a literature study focused on the selected target groups and health benefits. We conclude with some final remarks.

## 2. Defining the Potential Benefits of UGS

The first step of our approach consisted of a literature study to ascertain the possible health benefits of UGS. (We acknowledge that UGS may also pose threats to health, for instance, due to allergenic plant pollen (see, e.g., [19].) These and other threats should be carefully considered, e.g., in selecting vegetation.) Although we tried to avoid assuming a linear process of causal effects [18], following Lee et al. [8], we sought to identify the relationships between UGS features, the benefits they provide, and the activities they facilitate. To do so requires disentangling the various health benefits associated with UGS and understanding their mutual interdependencies. Let us illustrate with an example. Suppose the objective of an UGS intervention is to stimulate mental health. Physical activity is widely recognized to have a positive effect on mental health [20]. At the same time, physical activity also contributes to physical health, for example, by preventing cardiovascular disease [21]. On the other hand, just being outside in green spaces without performing any specific physical activity also improves poor mental health, especially when it is caused by stress [22].

Despite such difficulties, several authors have attempted to distinguish different health benefits of UGS. Zhou and Rana [23] have developed a framework for that purpose, identifying six “social” benefits: (1) providing recreational opportunities, (2) rendering aesthetic enjoyment, (3) promoting physical health, (4) adjusting psychological wellbeing, (5) enhancing social ties, and (6) providing educational opportunities. However, these benefits overlap to some extent and are thus not distinct; for instance, UGS that promotes recreational opportunities will often stimulate social contacts and thus contribute to the enhancement of social ties. Moreover, while some items on this list concern direct benefits, others concern opportunities which may or may not lead to benefits. Rakhshandehroo et al. [24], although employing different terminology, present a list of benefits similar to Zhou and Rana’s [23] that includes the confusing mix of benefits and opportunities for benefits. They add two more social benefits of UGS: crime reduction and accident reduction.

Thus, when creating design principles for UGS that promotes public health, which benefits should we focus on? The existing literature presents only limited guidance on this issue. We have selected three distinct health benefits that UGS can promote: (1) physical health, (2) social cohesion, and (3) psychological wellbeing (see Figure 1). We chose them for the following reasons: these benefits have already been addressed in the frameworks developed by other scholars and are well acknowledged in UGS studies; they are clear and distinct; they represent concrete measurable outcomes; and they reflect, or can directly be linked to, proven underlying mechanisms, namely physical activity, social interaction, and stress reduction (e.g., [4,8,17]). Figure 1 also depicts how these three benefits are affected by UGS use, that is, whether it is used by individuals or in groups: as Lee et al. [8] stress, the benefits of UGS stem not only from the characteristics of such spaces but also from their functionality—in other words, how and with whom UGS is used (e.g., alone or with others) influences its potential benefits. 

## 3. Baseline Study

The second step in our approach was to conduct a baseline study of Regenboogbuurt, which functions as our case study. This baseline study is needed to take the local context into account in our design principles, but also provides a “before” picture, which can, once the new designs have been implemented, be compared with an “after” study. 

Regenboogbuurt is a neighbourhood in Almere, The Netherlands, with approximately 5400 residents. It was developed in the 1990s as a tribute to the colourful work of its main source of inspiration, architect Bruno Taut. Colour, on the buildings and in the choice of vegetation, plays an essential role in its design, hence the district’s name, which translates into English as “Rainbow Quarter”. Two long avenues lined with ornamental Japanese cherries are references to Taut’s time in Japan. These trees, when combined with the houses behind them, produce the same colour effects as Taut’s urban design work in Berlin [25,26]. Such large-scale use of colour to structure urban space is unique in The Netherlands. Figure 2 and Figure 3 depict images typical of Regenboogbuurt’s design and include the cherry-lined avenues.

Regenboogbuurt’s greenery has been carefully selected to reinforce the colourful urban structure [27]. The neighbourhood is surrounded by green zones on three sides: a green boulevard to the north, a park to the west, and the banks of a canal to the south [27]. According to Pflug and Visser [27], the overarching design for its green space is based on the idea that there should be “homogeneous districts in a heterogeneous neighbourhood” in order to emphasize the differences between housing environments within the neighbourhood and to give specific meanings to different public spaces. As such, the neighbourhood is divided into four quadrants [27]. Our project focuses specifically on one of these quadrants, consisting of three connected green areas, as shown in Figure 4: two green “fingers” surrounded by houses (Numbers 1 and 2), separated by a rectangular pond in between (Number 3). The two green fingers are surrounded by pathways and are easily accessible. They both contain small playgrounds. The pond is lined with grass fields, a path, and some benches. It contains several islands planted with shrubs and trees, but they are not accessible. See Figure 5 and Figure 6 for images of the how the “green fingers” (Locations 1 and 2) look at ground level. We chose this quadrant because it is one of the last to be redeveloped in Regenboogbuurt, and thus we had ample time to develop design principles. Moreover, the socioeconomic status of the residents in this area is lower than in the rest of the neighbourhood, so the potential health benefits of green space are higher here [28].

The baseline study consisted of analyses of the demographics of the neighbourhood, or of the quadrant as far as data were available; residents’ health statuses; and their opinions on green space. We describe each of these issues below, after explaining the methods used to conduct the analyses.

### 3.1. Methods

We used several sources to analyse the demographics of Regenboogbuurt and the health of its inhabitants:The Extract of the Personal Records Database of Almere municipality (Uittreksel Basisregistratie Personen) and data from Allecijfers.nl (translated as “all numbers”: a website containing statistics on cities and neighborhoods, compiled from various studies published by the Central Bureau of Statistics (Centraal Bureau voor Statistiek–CBS) and other governmental agencies) were used to determine the number of inhabitants and their household compositions.Data from the Social Economic Planning Bureau (Sociaal Economisch Planbureau (SCP)) and the Extract of the Personal Records Database of Almere municipality were used to find information on income and socioeconomic status.Crime statistics were provided by the Almere Police Department.Information on the health status of adults was found through ‘Vraag Aanbod Analyse Monitor’ (VAAM, translated as “Supply and Demand Analysis Survey”) of Nivel (Dutch Institute for Research on Healthcare) (2011). Nivel conducted this survey with CBB (Cooperative Care in the Neighborhood) and ROS advisors (Regional Support Structures). The objective of the survey was to create a health risk analysis of Dutch neighbourhoods and present the number of visits to general practitioners for each postal code. VAAM was used to estimate the physical health risks present in Regenboogbuurt. We also used the Central Bureau of Statistics (2018) and the Health Survey of Municipal Health Services Flevoland (Gezondheidsmonitor by Gemeentelijke Gezondheidsdienst, GGD Flevoland) (2016). Flevoland is the province in which Almere is located. This health survey was created to illuminate the health aspects of Flevoland’s cities. It is based on a questionnaire that is sent out every four years. For residents under 19 years of age, we used the Youth Health Survey of Municipal Health Services Flevoland (2018), as Municipal Health Services Flevoland also sends out a questionnaire intended for minors. This questionnaire is less elaborate than the version for adults. Both surveys provide information about the municipality of Almere, not for Regenboogbuurt as such. We did check whether the statistics related to lifestyle and obesity found for Almere are consistent with the estimated health risks for Regenboogbuurt found through VAAM.

When possible, we compared the numbers found for Regenboogbuurt with those from two adjacent neighbourhoods (Bloemenbuurt and Indische buurt) to highlight Regenboogbuurt’s specificities. In some cases, we compared them with statistics for the city of Almere or The Netherlands as a whole. To better understand the neighbourhood, its residents, and the most pressing health problems they face, we also spoke to a city employee responsible for this neighbourhood who spends much time ‘on the street’, Mirjam Geurts, and to members of the Municipal Health Services.

We also explored residents’ opinions on Regenboogbuurt’s green spaces, as well as the ways in which they would like to see them improved. We used the results from an extensive survey conducted by BTL Advies, the company in charge of revitalizing Regenboogbuurt, in which 249 residents participated. Respondents were requested to assess the neighbourhood in general and its green spaces in particular. They were also asked about their dreams and wishes for the neighbourhood’s UGS [29].

Finally, we measured residents’ current use of green space to determine the functionality of the selected green spaces, that is, the activities they encourage, for instance, whether children use such spaces to play or whether people use them for jogging, and how much time people spend doing these activities. This is an essential part of the baseline study: current use can then be compared to that of the redesigned spaces. We used scan sampling and focal sampling to observe the use of the three selected green spaces. We performed these sampling measurements simultaneously, within observational periods of half an hour. Sampling was conducted between 9th September and 18th October, 2019, in all selected green spaces, three times a day for 16 days. The earliest measurements were taken at 8:49 a.m., and the latest at 3:53 p.m.; most were performed between 11:00 a.m. and 1:00 p.m. These measurements give only a picture of the number of people present in the three green spaces during these selected hours. Weather conditions were noted for each measurement.

*Scan sampling* provided us with a general overview of the number of people present in the selected space and the activities they performed. We divided each 30-minute measurement into 10-minute intervals: at T0 (at 0 min), T1 (at 10 min), T2 (at 20 min), and T3 (at 30 min), we noted the number of individuals and the activities they engaged in. Hence, we took 192 measurements for each selected green space and 576 measurements in total. *Focal sampling* contributed to our understanding of how much time people spend in the observed spaces and what they do during that time. During our observations, we selected one to three individuals to monitor more closely. We noted the activities they performed as well as the duration of these activities. Age and gender were estimated. Data were recorded using ethograms (see Table A1 and Table A2, Appendix A). We used the following method to select the individuals to observe: We did not observe people who were only passing through the area as they spent limited time in the space. Instead, we observed people engaged in activities for a longer period of time (such as playing in the playground or eating lunch). Because the number of people using the space was in general low, no more selection was needed.

### 3.2. Demographics

Regenboogbuurt is inhabited by many families with older children: 16.5% of residents are between 10 and 19 years old, and 21% are between 45 and 54. These figures are higher than in Almere overall (13.5% and 16.2%, respectively). At least one child lives in nearly half of all households, another figure also slightly higher than the Almere average (48.8% vs. 44.2%). The average yearly income in Regenboogbuurt is €23,800 a year (this includes people without wages: the average for working people is €31,700). This is €2000 lower than the average Dutch income and €1000 lower than the average Almere income, but €1500 higher than the average income in the adjacent neighbourhoods studied. Half the population is of Dutch origin, which is far below the Dutch average but common for Almere. The largest non-Dutch ethnicity here is Surinamese.

There are just over 2000 residential buildings in Regenboogbuurt: 72% are owner-occupied. The housing density is 19.3 dwellings per hectare, which corresponds to the Dutch average for cities of this size. Most houses (83%) are family houses and terraced. The quarter’s population is steadily growing, as the birth rate slightly exceeds the death rate. Compared to adjacent neighbourhoods and Almere as a whole, crime is low. The type of crime has changed over the years: shoplifting and substance abuse have diminished, but street robberies and delinquency have increased. Important services such as general practitioners, supermarkets, and schools are all available within a kilometre’s distance.

In sum, Regenboogbuurt’s socioeconomic status is lower than the Dutch average. It is a family-oriented neighbourhood (the 10–24 and 40–60 age groups are overrepresented) and has no particular crime-related problems.

### 3.3. The Health of Residents

The most common reason Regenboogbuurt inhabitants visit a general practitioner is cardiovascular disease. Fifteen percent of residents are obese; 53% are overweight. In Almere overall, 52% of residents between 19 and 34 years old, 53% of residents between 35 and 64 years old, and 26% of residents between 65 and 79 years old do not meet the Dutch Standard of Healthy Physical Activity (Nederlandse Norm Gezond Bewegen). Although these figures are below the national averages, ‘perceived health’ in Almere is comparable with Dutch figures (73% and 75% of people consider themselves to be in good health, respectively). The perceived health of Almere residents diminishes with increasing age; 83% of adults between 19 and 34 feel healthy, whereas only 43% of people over 80 years do so.

In terms of psychological wellbeing, loneliness is a serious problem in Almere (see Table 1). Almost 20% of Almere inhabitants face social exclusion, and 50% feel lonely at times. These figures are above the regional (Flevoland) and Dutch averages. While exact statistics about the psychological wellbeing of adults in Regenboogbuurt are not available, we do know that in Regenboogbuurt 13% of young people (<19 years old) suffer from anxiety or depression, 9% feel lonely, and 11% are bullied. Mirjam Geurts from the municipality also asserted that social cohesion in this neighbourhood is low.

In conclusion, the health issues present in Regenboogbuurt are comparable to major health issues in the rest of The Netherlands: physical activity is a point of attention (due to obesity and cardiovascular diseases), as is mental health, specifically loneliness.

### 3.4. Opinions on Green Space and Its Use

Almost three-quarters (74%) of people surveyed by BTL Advies [29] like living in Regenboogbuurt. Most people are satisfied with the quantity of green space, its quietness, and the (colourful) trees. They are less content with the quality of Regenboogbuurt’s green space, complaining specifically about poor maintenance and about people parking their cars in green spaces. Moreover, they consider both the quantity and the quality of sport and playground facilities to be too low. Residents living close to our studied green spaces contend that the number of playgrounds is sufficient, but that they are far apart, not always easily or safely accessible, and of poor quality. Spaces for recreational or social activities (e.g., meeting spaces) are lacking or of bad quality. Families with children in particular judge the quality of green areas and playgrounds to be below standard. Residents indicate a particular need for (natural) children’s playgrounds and walking trails with seating facilities [29].

During our 576 observations (over 16 days, three measuring moments per day, each consisting of four measurements, in three locations), we noted 213 individuals using the three locations we studied (see Figure 4). (Because we conducted our observations in succession and the three selected green spaces are close to each other, it is possible that someone was first counted in one location and later in another location). The most common activities people performed were walking, dog walking, and playing on the playground. People spent on average 10 min in these green spaces. Figure 7, Figure 8 and Figure 9 show that all three locations were mostly found to be empty during scan sampling: The figures show the number of people observed during the 192 measurements in each of the three green spaces. 

Location 1 consists of two meadows with a playground in the middle (with swings, seesaws, a climbing frame, and a carrousel). It was busiest in the morning; after 1:00 p.m. this space was hardly used. The meadow to the north of the playground was not used at all. Activities performed here were walking and using the playground.Location 2 looks similar to Location 1: It also consists of two meadows with a playground in between. This playground is larger and differently designed. There is a small concrete football pitch, a baby swing, a seesaw, and a slide. Activities performed here are similar to those at Location 1: People mostly walked or used the playground. The meadows were barely used: The southern meadow was not used at all. One group of people ate lunch at a picnic table here.Location 3 was the most used of all three areas. It consists of a pond with paths and grass around it. The pond contains several small islands, hosting birds. It was mostly visited by women, who often strolled or walked their dogs. They walked not only to get from place to place, but also for recreational purposes, and sat on the benches to look at the water.

For background data for Figure 7, Figure 8 and Figure 9, see Table A3 (Appendix B).

Hence, although Regenboogbuurt features several green areas, they are not generally perceived to be particularly inviting [29]. Our finding that all three locations were empty most of the time confirms this observation.

## 4. Selecting Health Benefits and Target Groups

The third step in our approach was to select one or more health benefits and one or more target groups to focus on. In keeping with Webb et al.’s [11] emphasis on local contexts, we aimed to concentrate on what is most needed in the area in terms of demographics and health issues.

We encountered some difficulties in the selection process. First, the numbers from the neighbourhood analysis were not easily comparable: different agencies worked with different age categories, and data were not always available at the neighbourhood scale or were not available at all. Second, it was difficult to conclude what the pressing health issues were. Should we be looking at the neighbourhood itself, or should we compare it to other neighbourhoods? In other words, is it better to focus on absolute numbers (i.e., reaching a larger group) or on relative numbers (i.e., targeting a specific problem)? Third, we had to consider several pragmatic issues because our future design principles are intended to be implemented in the real world. For instance, we had to take the goals of the revitalization company into account, as it would be the one incorporating the design principles.

The selection process was an iterative process in which we considered three potential health benefits and several potential target groups. The potential health benefits of UGS are important for all target groups and are interrelated, as we have shown. However, to provide evidence that specific UGS designs can promote health, we felt the need to target both user age and benefits as specifically as possible.

Based on the neighbourhood’s characteristics, we decided to focus on designs that promote physical activity for youth and adolescents aged 10–24, and social cohesion for adults aged 40–60. Our decision was based on the following reasons:The 10–24 and 40–60 age groups are overrepresented in Regenboogbuurt. (The 60+ age group is highly underrepresented, the 25–40 age is underrepresented, and the under 10 age group is proportionally represented compared with other neighbourhoods).Almost half the residents in the youth/adolescent (10–24) age category do not meet the Dutch Standard for Healthy Physical Activity; 26% of young people have a BMI > 25. While physical activity levels are low for most age groups, the literature reveals that youth is particularly prone to physical inactivity, which declines even more during the transition from childhood to adolescence [31]. Also, physical activity in adolescence appears to be an indicator of physical activity in adulthood [31,32].Loneliness is a major problem amongst middle-aged people (40–60). Loneliness is, moreover, a key issue in The Netherlands more broadly: 80% of Dutch people consider it to be a serious problem [33]. Figures for Regenboogbuurt are eight percentage points above the Dutch average (62% and 54%, respectively, are very lonely).The UGS is of bad quality and is hardly used. People confirmed this in the survey conducted by BTL Advies and suggested more playgrounds and sports facilities be built.

## 5. Consequences for Urban Green Space Design

Thus far, we have disentangled the health benefits associated with UGS, analysed the neighbourhood and its residents, and decided to focus on two specific age groups and support a specific health benefit that UGS can provide for each group. Our final step was to understand what UGS designs can promote these benefits, and therefore, we went back to the literature. As literature focusing on what elements of UGS design lead to either physical activity in youth/adolescents or social cohesion in middle-aged people is scarce, in the following two sections we start by describing findings from the literature on urban space in general before going into the literature on urban green space in particular, if it is available at all. In some cases, we included findings on other (or broader) age groups.

### 5.1. Physical Activity and Youth/Adolescents

Sivam et al. [34] present a literature review that lists urban design principles that encourage physical activity (walking, cycling, exercise), and argue that the following attributes of UGS should be taken into account in the design process: permeability, diversity or richness, and spatial openness. Huston et al. [35] argue that fostering street connectivity is amongst the most important design principles, stating that it is more important for older children than younger ones. Connectivity is critical for being able “to go somewhere”: to be active or meet peers. Especially for those over 18 years of age, places explicitly set aside for physical activity are important [35]. Places for recreational opportunities are crucial as well (younger children: [36]; adolescents: [31,37]), but these must be within walking distance (adolescents: [38]). “Richness” does not necessarily mean “diverse” [39] and “dense”; research on public space shows that lawns; sports fields; and open, adventurous playgrounds encourage physical activity [40]. Playground design is also associated with physical activity levels [41].

These design principles can be applied outside of UGS, but that would not take the benefits of *green* space into consideration. Indeed, Gardsfjord et al. [32] note that it is worth applying these ideas in UGS design; for young people, access to green space is critically important for promoting physical activity. Other studies have also demonstrated that children and young people prefer “naturalness”, whether in the form of trees or flowerbeds [40,42].

Several studies have also pointed out that social factors play a role in enhancing physical activity in UGS, suggesting that urban spaces that encourage social interaction also stimulate physical activity (see e.g., [43]). Van Hecke et al. [31] have found that when visiting parks, youngsters prefer the presence of their peers, whether they are active or inactive, although when they visit a park with the specific aim of exercising, they prefer a park with active or no peers over a park with inactive peers. Thus, creating opportunities for adolescents to be active together seems important.

Finally, both Gardsjord et al. [32] and Moore et al. [9] have addressed gender differences regarding urban space and physical activity: some studies have found girls to be less active than boys, partly due to perceived gender stereotypes (e.g., football is for boys). In contrast, Akpinar [42] found a positive correlation between the existence of sports fields and physical activity amongst girls.

### 5.2. Social Cohesion and Middle-Aged People

Whereas scientific knowledge about the positive association between UGS and social cohesion in general is abundant, convincing, and growing [4,44], little is known about which characteristics of UGS stimulate social cohesion for specific target groups. Therefore, we drew upon studies that were less age specific. Research on urban space in general shows that open spaces can enhance social cohesion [45], but it is the general infrastructure and the spatial elements that matter most. Social interactions occur more often in circulation spaces, and less often in seating spaces or spaces without a clear function. One study that examined design elements that increase social interaction found the presence of specified routes and nodes most important [46]. Other studies have shown that street furniture encourages social interaction [47,48].

In their study of UGS, Rasidi et al. [49] found that, besides paved areas and basketball courts, less densely vegetated and more open areas with undulating landforms were favoured places to meet. However, they determined no particular characteristics of UGS that specifically support the use of these spaces: only diversity of UGS mattered. Moreover, the extent to which the design of UGS stimulates social cohesion is culture dependent: Peters et al. [50] found Dutch park visitors to be active (cycling, walking), while non-Western immigrants in the same parks would meet and sit together with family. Finally, integrating spatial and design elements can foster certain social behaviours. For instance, concave seating areas next to plants, close to activity spaces, and with water features or artworks within sight can provide users of such seats with shade, events to watch, and aesthetic experiences, and hence enable interactions [46].

### 5.3. UGS Design Principles for Regenboogbuurt

The objective of this study was to formulate UGS design principles for one part of the Regenboogbuurt neighbourhood in Almere, The Netherlands. We conducted a neighbourhood analysis—in which we examined residents’ socioeconomic standings, health statuses, and opinions on UGS and its use—to choose our target groups: youth/adolescents (10–24) and middle-aged people (40–60).

Based on our interpretation of the literature, we argue that to stimulate physical activity amongst youth/adolescents and to support a diversity of activities, UGS design has to include a variety of easily accessible natural amenities, facilities, and open spaces. It should also improve the neighbourhood’s permeability. UGS should stimulate social interactions that in turn increase physical activity. Following these design principles would satisfy the needs of residents who have indicated that both the quantity and the quality of sports and playing facilities are insufficient.

Social cohesion amongst middle-aged residents could be stimulated by a slightly undulating landscape with a mix of open spaces and subspaces, connected by an interesting network of paths and nodes. In such subspaces, concave seating furniture located in natural, aesthetically interesting environments close to water or artworks has the potential to stimulate social interactions. The resulting diversity can satisfy the needs of the different cultures that use urban parks and natural areas, as found by Peters et al. [50].

These UGS design principles were shared with Ms Rommers, BTL’s landscape architect, who is responsible for designing Regenboogbuurt’s UGS. She saw possibilities for applying our design principles either within the green areas we studied or nearby. She confirmed the importance of green spaces for young people’s social interactions and recognized the potential to create meeting spaces for both young and middle-aged people around the square pond.

## 6. Conclusions

By describing our approach to developing design principles for our case study in Regenboogbuurt, we aimed to outline a methodology that can be used to create a set of design principles that takes the unique features of a particular neighbourhood into account. We conclude that to stimulate physical activity amongst youth and social cohesion amongst the middle-aged, the UGS in Regenboogbuurt must feature diversity, in terms of natural amenities, facilities, and both open and more closed spaces. It should contain walking paths, places to play sports, and interesting places to linger, and connect the neighbourhood with other urban places.

As the design principles we developed have yet to be implemented, the results of applying them remain unknown, and thus, so too does the effectiveness and universality of our approach. Nevertheless, we aim to verify its success later. Meanwhile, we hope to have inspired others with this account of our approach to taking academic findings into account in a real-life situation—including the struggles we encountered on the way. The strength of our approach lies in its extensiveness: we spent considerable time understanding how health benefits are related and mapping the neighbourhood in detail. We used the knowledge we acquired to make an informed decision about a design targeted to provide health benefits to specific demographic groups. Afterwards, we studied the literature to understand which design elements might stimulate these benefits. Moreover, our neighbourhood mapping exercise can be considered a “before study”. This approach, or methodology, therefore helped us create the conditions that later on—after the revitalization of Regenboogbuurt is finished—will allow us to determine to what extent our design principles have had the expected results. As such, it is a first step towards more evidence-based UGS design. We invite others to use, evaluate, and improve our methodology. We suggest others perform sampling not only during workdays but also in the evenings and at weekends to get a more accurate picture of UGS use.

Finally, whether designs have their intended outcomes also depends on urban governments. Designers can come up with stimulating designs, but policymakers need to make sure that UGS is maintained and safe so that people use it. As Webb et al. [11] (p.69) argue, urban futures are emergent and driven by multiple drivers that are not amenable to planned solutions. Moreover, several of these drivers are interconnected and while they can be influenced locally, they are beyond the control of any single jurisdiction. We therefore call upon local policymakers from various disciplines to recognize the value of UGS, to create conditions for UGS to be inviting for urban residents, and as such, to increase the extent to which UGS design will live up to its potential to promote health. Involving residents in the final designs will also be key to encouraging the use of UGS.

## Figures and Tables

**Figure 1 ijerph-17-05205-f001:**
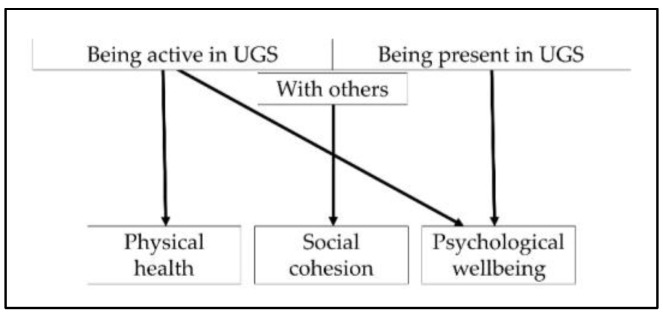
Model for assessing the health benefits of UGS and how they relate to the way it is used.

**Figure 2 ijerph-17-05205-f002:**
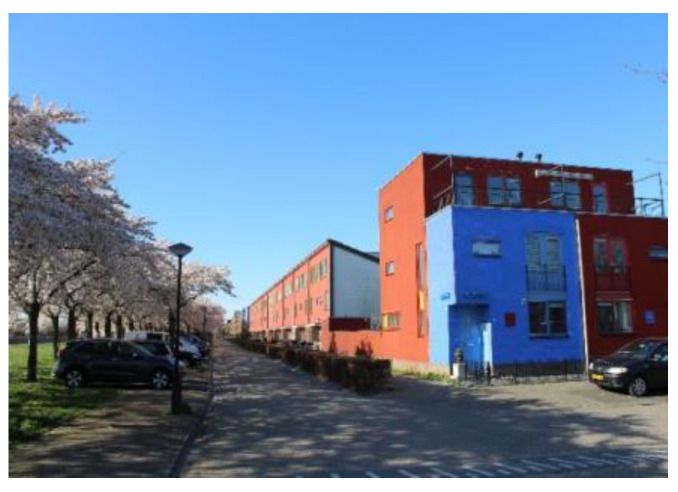
Cherry-lined avenue with red and blue houses (courtesy of Author 2).

**Figure 3 ijerph-17-05205-f003:**
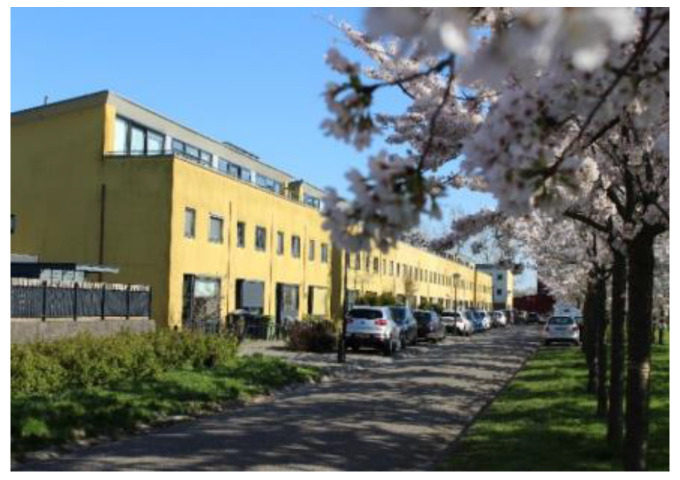
Cherry-lined avenue with yellow houses (courtesy of Author 2).

**Figure 4 ijerph-17-05205-f004:**
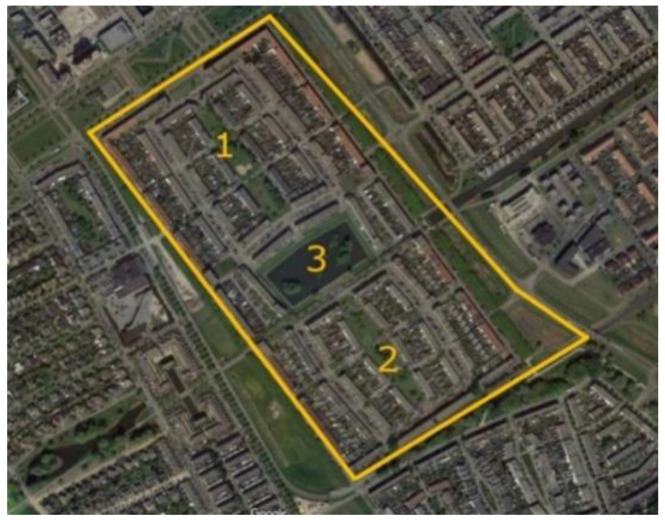
The three green spaces on which our project focuses (Google Maps).

**Figure 5 ijerph-17-05205-f005:**
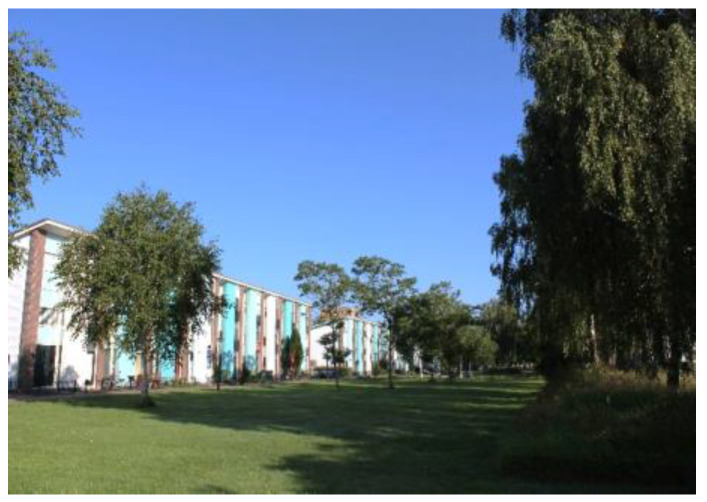
Green space in Regenboogbuurt: Location 1 (courtesy of Author 2).

**Figure 6 ijerph-17-05205-f006:**
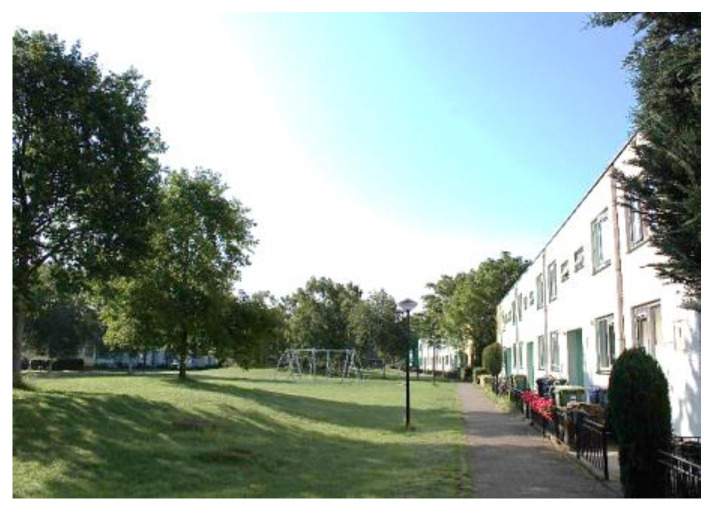
Green space in Regenboogbuurt: Location 2 (courtesy of Author 2).

**Figure 7 ijerph-17-05205-f007:**
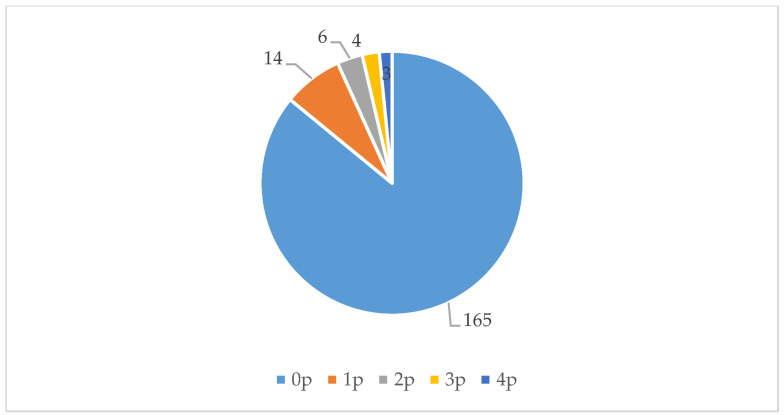
The number of people (p) observed during each measurement at Location 1.

**Figure 8 ijerph-17-05205-f008:**
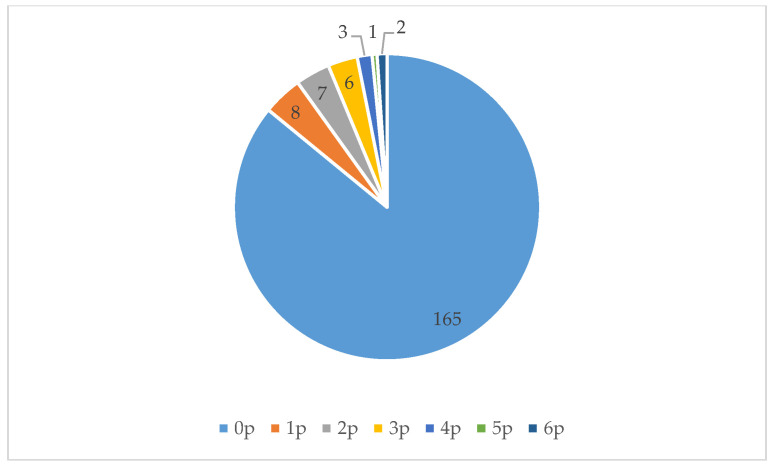
The number of people (p) observed during each measurement at Location 2.

**Figure 9 ijerph-17-05205-f009:**
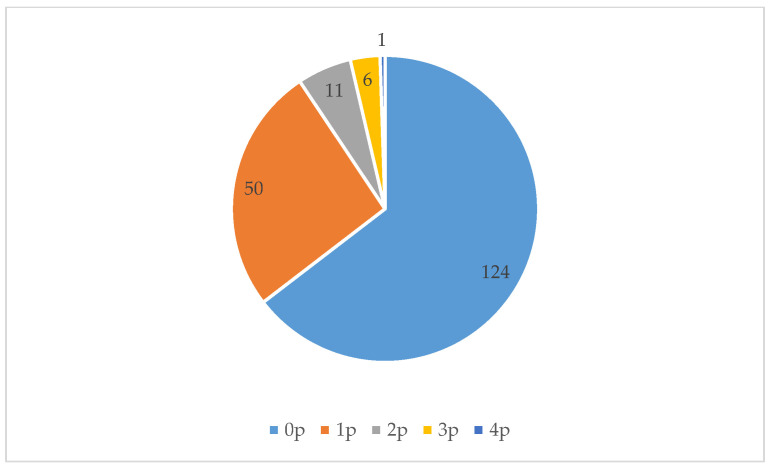
The number of people observed during each measurement at location 3.

**Table 1 ijerph-17-05205-t001:** Loneliness in Almere (2016) [30].

	19–34 Years Old	35–64 Years Old	65–79 Years Old	>80 Years Old
**Not lonely**	65%	52%	55%	41%
**Somewhat lonely**	27%	35%	36%	48%
**Very lonely**	5%	8%	6%	8%
**Extremely lonely**	3%	5%	3%	3%

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
