# Peer review of "Designing Urban Green Space (UGS) to Enhance Health: A Methodology"

_ijerph, 2020, doi:10.3390/ijerph17145205_

Round 1
Reviewer 1 Report
The authors proposed an interesting approach to the design of green areas in cities, taking into account their impact on public health. They have made a broad analysis of the factors that should be considered when designing green areas. A certain lack is the omission of threats posed by badly designed green areas. Especially important in recent years are the risks of allergenic pollen that are emitted by green areas. To design well-friendly green areas for residents, you need to know what species of plants and trees are the largest producers of allergenic pollen. You also need to know if the risk is the same in all green areas (city forest, park) - see the work by Dudek et al. 2018: https://doi.org/10.1007/s10342-018-1144-x
Pictures of greenery from the period when the trees have no leaves do not reflect the proper structure of this greenery in the space of a given landscape interior (Figs. 5 and 6).
Observing the use of urban green areas from 8:49 AM to 3:53 PM may not give a proper full picture. Because working people often visit green areas after work - in the afternoon. Scan sampling does not give a proper picture as to the number of people present in a selected space during the day but only at selected hours. This should be emphasized.
What is shown in Figures 7-9? Figures are badly signed: this is not the number of people present in a given location - the sum is not 213. Maybe this is the average (out of 192 measurements) number of people staying in a given location during a single measurement?
You should add data labels in percentages ?? participation of individual groups: 1 person, 2 people etc.
The final remarks lack emphasis in the points of the developed design principles - this needs to be supplemented.
The concrete results of demographics and problems of individual age groups presented in the paper are primarily of local importance. However, the methodical approach to the problem of designing green areas in cities can also be used elsewhere and this is the highest value of this research.
Reviewer 2 Report
The framework of this manuscript is circumlocutory without following the regular structure of this journal (e.g. introduction, methods, results, discussion, conclusion, etc.), which makes it reads like a report rather than an academic paper. For an original research, pertinent literature review would better be within the “introduction” as the theoretical basis of this study or present a research gap.
This manuscript seems to be between the “original research” and “review article”, for the assessment model and methodology generated are almost based on previous literature, which essentially are the summary and generalization of existing research results, while case study is analyzed at the same time. Even though the assessment model and methodology can be theoretical contributions, they need to be better related to the case study. Specifically speaking, without case study in this manuscript, theoretical contributions can also be concluded. Most importantly, the result of neighborhood revitalization by using the methodology generated is still unknown, which can’t verify the efficiency and universality of the methodology.
This study is kind of unfinished. If there can be any improvement, the authors may transform this manuscript into a review article, focusing on “what UGS criteria offer what particular benefit for what target group”, and developing a more accurate assessment model and methodology based on previous researches. After the neighborhood revitalization is done in the future, an original research can be generated using the baseline study in this manuscript, the derived design principles from the review article as well as the performance of revitalization to verify the efficiency and universality of these principles.
Reviewer 3 Report
This paper challenged to descibe the relationship between Urban Green Space (UGS) and health.
Its appricahes are quite unique and it showed very interesting results, too.
Especially, surveying results about the lonliness of the residents are excellent!
If you can add a little bit more, please describe the architectute conditions, too.
You showed some architectures on Fig.1.
The buildings are reinforced concrete, however their walls are colorful.
In addition, you should explain how the buildings occupied the residential space (height and width).
Round 2
Reviewer 2 Report
the authors have responded to most of my comments well, although some issues relevant to research design and structure has been innate.
My suggestions are, making the logic threads more fluent, and English proofreading. The current version is kind of transcript of oral presentation.
Author Response
Dear reviewer,
Thanks for taking the time to look at our improved manuscript and to review it again.
We took your advice at heart and had the manuscript proofread for English. Our proofreader has been very thorough, and he helped us be more clear in the 'logic thread' as well. He had a number of clarifying questions. We tackled each one carefully, so that we are confident that the manuscript is more readable and that the thread of the story is better understandable.
As we made numerous small changes and have tried to clarify the logic thread also with small changes here and there, it is difficult to be precise about what we have changed overall. However, the main changes are to be found in lines 57 to 77, and in the section final remarks (second paragraph). We also changed the wording 'methodology' into 'approach', to be more clear about what it is we are doing in the manuscript.
We hope that we solved your concerns with the manuscript, and thanks again for taking the time to help us with this work,
The authors